# The dynamic nature of the human origin recognition complex revealed through five cryoEM structures

Matt J Jaremko[1,2,3†], Kin Fan On[1,2,3†], Dennis R Thomas[1,3], Bruce Stillman[3], Leemor Joshua-Tor[1,2,3]*

[1]W. M. Keck Structural Biology Laboratory, New York, United States; [2]Howard Hughes Medical Institute, New York, United States; [3]Cold Spring Harbor Laboratory, New York, United States

**Abstract** Genome replication is initiated from specific origin sites established by dynamic events. The Origin Recognition Complex (ORC) is necessary for orchestrating the initiation process by binding to origin DNA, recruiting CDC6, and assembling the MCM replicative helicase on DNA. Here we report five cryoEM structures of the human ORC (HsORC) that illustrate the native flexibility of the complex. The absence of ORC1 revealed a compact, stable complex of ORC2-5. Introduction of ORC1 opens the complex into several dynamic conformations. Two structures revealed dynamic movements of the ORC1 AAA+ and ORC2 winged-helix domains that likely impact DNA incorporation into the ORC core. Additional twist and pinch motions were observed in an open ORC conformation revealing a hinge at the ORC5·ORC3 interface that may facilitate ORC binding to DNA. Finally, a structure of ORC was determined with endogenous DNA bound in the core revealing important differences between human and yeast origin recognition.

*For correspondence: leemor@cshl.edu

†These authors contributed equally to this work

## Introduction

DNA replication is essential to all forms of life. In eukaryotes, replication is initiated from multiple start sites on chromosomes called replication origins (reviewed in *Bell and Labib, 2016*; *Bleichert et al., 2017*; *Leonard and Méchali, 2013*; *On et al., 2018*; *Prioleau and MacAlpine, 2016*). The very first step of this initiation process is accomplished by DNA association with the Origin Recognition Complex (ORC), a six-subunit protein that forms a partial ring around origin DNA (*Li et al., 2018*; *Yuan et al., 2017*). In *Saccharomyces cerevisiae*, this is a sequence-specific binding event, however, in metazoans the manner in which origins are identified is not yet clear. The bromo-adjacent homology (BAH)-domain at the N-terminus of ORC1 facilitates binding of ORC to DNA through nucleosomes which contain specific histone modifications that influence ORC1 affinity to the chromatin complex (*Eaton et al., 2010*; *Hossain and Stillman, 2016*; *Kuo et al., 2012*; *Müller et al., 2010*; *Noguchi et al., 2006*; *Rivera et al., 2014*). Once bound to origins, the ORC-related AAA+ ATPase CDC6 enters the complex to complete the ORC-CDC6 ring around the DNA (*Yuan et al., 2017*). Subsequent recruitment of the licensing factor CDT1 bound to the replicative helicase MCM hexamer complex then leads to the formation of MCM double-hexamers assembled around dsDNA in a head-to-head association (*Evrin et al., 2009*; *Miller et al., 2019*; *Remus et al., 2009*). During origin licensing, ORC must perform several tasks sequentially: bind DNA, find origin sites, and dissociate from DNA upon completion of the MCM double-hexamer loading and assembly. In fact, single molecule studies have shown that ORC is released from DNA immediately following stable association of two MCM molecules with the DNA substrate (*Ticau et al., 2015*). These dynamic events likely require significant conformational changes in ORC.

High resolution structures of ORC from *Saccharomyces cerevisiae* (ScORC), *Drosophila melanogaster* (DmORC), and *Homo sapiens* (HsORC) have revealed the detailed architecture of the complex (*Bleichert et al., 2015*; *Li et al., 2018*; *Tocilj et al., 2017*; *Yuan et al., 2017*). The complex consists of six subunits (ORC1-6), with subunits ORC1-5 forming a partial ring capable of encircling DNA (*Li et al., 2018*; *Tocilj et al., 2017*; *Yuan et al., 2017*). All five subunits contain a winged-helix domain (WHD) and a AAA+ domain which consists of the RecA-fold and lid domains (*Figure 1a*), although the ORC2 and ORC3 RecA-folds have diverged slightly and do not contain a lid domain. Adjacent ORC subunits associate in a double-layer arrangement through interfaces between the WHDs in one layer and the AAA+ domains in another with ATP-binding sites sandwiched between the ORC1·4, ORC4·5, and ORC5·3 AAA+ interfaces (*Figure 1g*; *Tocilj et al., 2017*). The ORC1·4 and ORC4·5 sites represent canonical AAA+ ATPase sites (*Enemark and Joshua-Tor, 2008*; *Erzberger and Berger, 2006*), while the ORC5·3 interface is considerably more flexible relative to the other ATP interfaces. Of the three ATP-binding sites, only the ORC1·4 site exhibits ATPase activity and this activity is essential for cell viability (*Bowers et al., 2004*; *Klemm et al., 1997*; *Randell et al., 2006*; *Tocilj et al., 2017*). In addition, several mutations at the ORC1·4 ATPase site are associated with the developmental disorder Meier-Gorlin Syndrome (*Bicknell et al., 2011a*; *Bicknell et al., 2011b*; *de Munnik et al., 2012a*; *de Munnik et al., 2012b*; *Guernsey et al., 2011*). However, the functional role for ATP hydrolysis is still unknown.

Based on the previous structures, ORC exhibits two main conformations differing primarily in the positions of the ORC1 AAA+- and ORC2 WH-domains: an open state (ScORC and HsORC) (*Li et al., 2018*; *Tocilj et al., 2017*; *Yuan et al., 2017*) and a closed, auto-inhibited state (DmORC) (*Bleichert et al., 2015*). In the open state, the ORC1 AAA+-domain binds to the ORC4 AAA+-fold to form the ORC1·4 ATPase active site and ORC2 WHD is adjacent to the ORC3 WHD. The ring-shaped core of ORC in the open state is accessible to solvent and DNA substrates. In the closed state, the ORC1 AAA+ motif is dramatically rotated by ~80° towards the ORC2 WHD position and ORC2 WHD is collapsed into the open ring-shaped core of the complex. The collapsed ORC1 AAA+-domain and ORC2 WHD in the closed state completely obstruct the core of ORC and could not accommodate DNA. In addition, the ORC1·4 AAA+ interface is disrupted and, therefore, the main ATPase site is disrupted. Initially, the closed state was only seen in DmORC, but recent cryoEM studies on HsORC suggested that the ORC1 AAA+ domain also adopts the auto-inhibited conformation (*Bleichert et al., 2018*). Further structural analysis is necessary to determine the prevalence and function of the auto-inhibited state in other species.

As mentioned, origin recognition, which is conferred by ORC binding to DNA, differs between yeast and metazoans. In the yeast *Saccharomyces cerevisiae*, ORC specifically recognizes and binds to a class of DNA sequences, called the autonomously replicating sequences (ARSs), which act as the replication initiation sites in vivo (*Bell and Stillman, 1992*; *Brewer and Fangman, 1987*; *Eaton et al., 2010*; *Huberman, 1987*; *Linskens and Huberman, 1988*; *Marahrens and Stillman, 1992*; *Rao and Stillman, 1995*; *Theis et al., 1999*). Previous ScORC structures revealed multiple regions of Orc1, Orc2, Orc4 and Orc5 that interact with double stranded DNA, including a lysine-rich region in an unstructured domain of Orc1, as well as two motifs within the AAA+ domain in each subunit that interact with elements of ARS DNA: the initiation-specific motifs (ISM) in the AAA+-domains and β-hairpin wings in the WHDs (*Bleichert et al., 2017*; *Li et al., 2018*; *On et al., 2018*; *Yuan et al., 2017*). Of particular importance are the ISM of Orc2 and an α-helix inserted into the β-hairpin wing of Orc4. The Orc2 ISM binds to the DNA minor groove with the W396 sidechain tucked into the hydrophobic minor groove at the T-rich region within the ARS consensus DNA sequence (*Li et al., 2018*). Interestingly, the α-helix insertion in the β-hairpin wing of Orc4 extends deep into the DNA major groove and likely plays an important role in DNA sequence-specific interactions with the ARS sequence (*Hu et al., 2020*). The β-hairpin wing of Orc4 is unresolved in the DmORC and HsORC structures and the importance of this region is unknown in metazoans. In metazoans, notably, ORC lacks sequence specificity for origin recognition and may rely on interactions with nucleosomes to establish DNA origin sites (*Noguchi et al., 2006*; *Remus et al., 2004*; *Vashee et al., 2003*).

In this study, we have determined cryoEM structures of the human Origin Recognition Complex in various conformational states and at higher resolution than previous studies. Four cryoEM maps revealed dynamic movements of ORC without DNA. The ORC1 AAA+ domain and the ORC2 WHD seem to repel each other to create dynamic movement between two extreme states. In addition, a

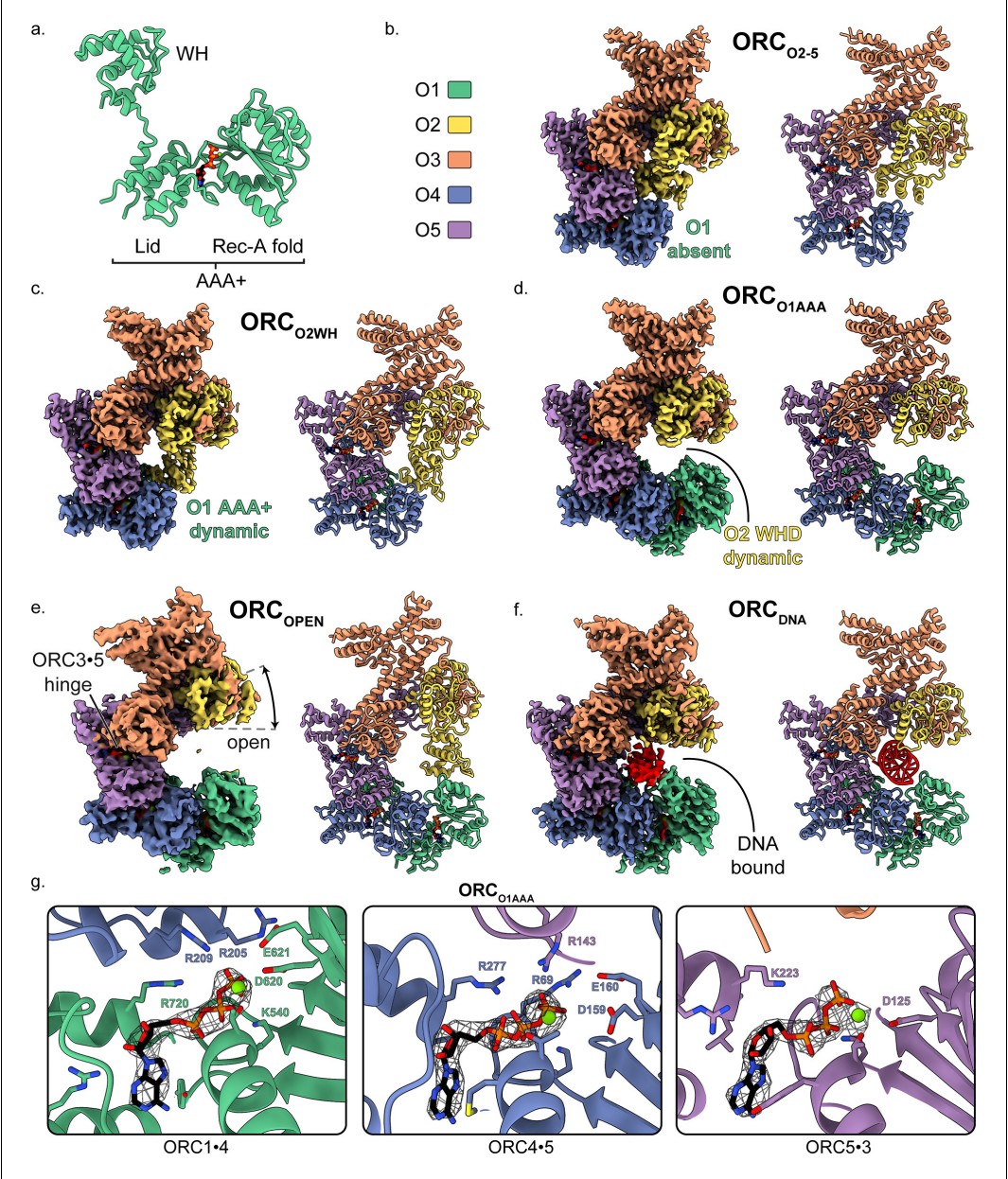

**Figure 1.** Overall architecture of human ORC. (a) Cartoon representation of the general structure of an ORC subunit with a winged-helix [WH], and a AAA+ domain containing a lid and RecA-fold, as illustrated by ORC1 from ORC$_{O1AAA}$. Human ORC structures were determined in five conformations: (b) ORC$_{2-5}$, (c) ORC$_{O2WH}$, (d) ORC$_{O1AAA}$, (e) ORC$_{OPEN}$, and (f) ORC$_{DNA}$. Each panel contains a density map and model and is color coded by subunit as illustrated in panel (b). ATP carbon atoms are colored in black and the remaining atoms, including Mg$^{2+}$ (green spheres), are colored by the CPK and Jmol element colors. (g) Close-up of the three ORC ATP sites in ORC$_{O1AAA}$ with density maps overlaying the model. Residues important for coordination of the ATP molecules and Mg$^{2+}$ are labelled. DNA is colored red/grey and all displayed atoms are colored by the CPK and Jmol element colors.

The online version of this article includes the following figure supplement(s) for figure 1:

**Figure supplement 1.** ATP sites of each ORC conformation.
**Figure supplement 2.** CryoEM workflow of ORC1-5.
**Figure supplement 3.** CryoEM workflow of ORC2-5.
**Figure supplement 4.** CryoEM map resolutions of ORC1-5.
**Figure supplement 5.** CryoEM map resolution of ORC2-5.
**Figure supplement 6.** 3DFSC analysis of the ORC conformations.
**Figure supplement 7.** Resolution improvement of ORC2 and ORC3 near the ORC gap opening of ORC$_{O1AAA}$.
**Figure supplement 8.** The intertwine between ORC subunits 2 and 3.

hinge was identified at the ORC3·5 interface that creates prominent twist and pinch movements of the entire complex. Lastly, a structure of ORC bound to co-purified, endogenous DNA (ORC$_{DNA}$) was determined that revealed several ORC regions in close contact with DNA. The ORC2 WH- and ORC1 AAA+-domains are both visible in this ORC conformation and bind DNA directly. The various ORC conformations provide insight into the dynamics of ORC during DNA replication initiation.

## Results and discussion

### Overall HsORC structure and comparison to ORC from *Saccharomyces cerevisiae* and *Drosophila melanogaster*

Two HsORC constructs were generated for recombinant expression based on our previous study (*Tocilj et al., 2017*) with the following changes: ORC2 was extended to full-length, the StrepTag was moved from the N-terminus of ORC1 to the N-terminus of ORC3 to improve sample purity during affinity purification, and the Sumo tag on ORC1 was removed. One construct consisted of full-length ORC1, but after expression and purification, ORC1 was absent yielding a complex of ORC2-5 (henceforth labelled ORC$_{O2-5}$). Another construct included a truncated version of ORC1 [aa471-861] that yielded a purified complex of ORC1-5 with multiple conformations. We utilized cryo-electron microscopy (cryoEM) to investigate ORC due to the various conformations and potential flexibility seen in previous ORC structures (*Bleichert et al., 2015*; *Li et al., 2018*; *Tocilj et al., 2017*; *Yuan et al., 2017*). The cryoEM analysis lead to a 3.5 Å structure of the ORC2-5 complex (ORC$_{O2-5}$) and four distinct structures of the ORC1-5 complex: ORC$_{O1AAA}$ (3.2 Å), ORC$_{O2WH}$ (3.7 Å), ORC$_{OPEN}$ (4.0 Å), and ORC$_{DNA}$ (4.3 Å) (*Figure 1—figure supplement 1–4* and *Supplementary file 1*). The four structures determined from ORC1-5 illustrate the flexibility and dynamic movements of ORC when ORC1 is present in the complex.

The ORC$_{O2-5}$ structure that lacks the ORC1 subunit consists of subunits ORC2-5 in a compact and substantially closed conformation (*Figure 1b*). The ORC2 WHD of ORC$_{O2-5}$ is collapsed into the ring-shaped core of the complex with the β-hairpin wing facing the ORC4 WHD (*Figure 2c*). The ORC2 WHD is significantly deeper in the core and rotated ~135° compared to the auto-inhibited state of ORC2 WHD in the DmORC structure (*Figure 2d* and *Video 1*; *Bleichert et al., 2015*). Once ORC1 is bound to form ORC1-5, the complex opens up into several related conformations to accommodate ORC1 (*Figure 1c–f*). The first of these, ORC$_{O2WH}$, consists of a dynamic ORC1 AAA+ domain that is unresolved in the refined map (*Figure 1c*). Interestingly, in this structure, the ORC2 WHD was stably bound inside the ring-shaped core of ORC in a conformation that is similar to the auto-inhibited state observed in the DmORC structure (*Figure 2d,f*; *Bleichert et al., 2015*). In contrast to ORC$_{O2WH}$, ORC$_{O1AAA}$ consists of a dynamic ORC2 WHD that is unresolved in the refined map (*Figure 1d*). The ORC1 AAA+ domain in this structure is ordered and forms the important interface with ORC4 with an ATP coordinated between the two subunits, an ATPase active conformation seen in previous ScORC and HsORC structures (*Li et al., 2018*; *Tocilj et al., 2017*; *Yuan et al., 2017*). The flexibility of ORC2 WHD is also observed in ScORC and in the DNA-bound ScORC, where the domain was located in several states proximal to DNA (*Li et al., 2018*), and different from its position in the yeast OCCM structure (*Yuan et al., 2017*). Thus, the dynamic movements of the domain are prevalent across species and likely influence binding to DNA and CDC6. The subunits of the third conformational state of the ORC1-5 structure, ORC$_{OPEN}$, are arranged in a more open configuration relative to the previous two structures where the overall spiral of the complex is widened (*Figure 1e* and *Figure 3a*). The conformation of ORC$_{OPEN}$ resembles the ScORC conformation bound to Cdc6 (*Yuan et al., 2017*). It was noted previously from low resolution cryo-EM structures comparing ScORC on DNA with ScORC-Cdc6 on DNA that substantial changes in the location of parts of the Orc1 subunit occurred upon binding of Cdc6 to ORC (*Sun et al., 2012*). Finally, the cryoEM map of ORC$_{DNA}$ consists of DNA bound in the core of the complex (ORC1-5) with several ORC subunits making contacts to the DNA (*Figure 1f*, *Figure 4*). The ORC1 AAA+ and ORC2 WHD of ORC$_{DNA}$ most likely envelop DNA predominantly through electrostatic interactions. The ORC$_{OPEN}$ and ORC$_{DNA}$ maps suffered from the existence of a preferred orientation which likely contributed to the lower resolutions of these maps (*Figure 1—figure supplements 4* and *6*). In addition, the nominal resolution of these maps, determined from their optimal direction is higher than their overall

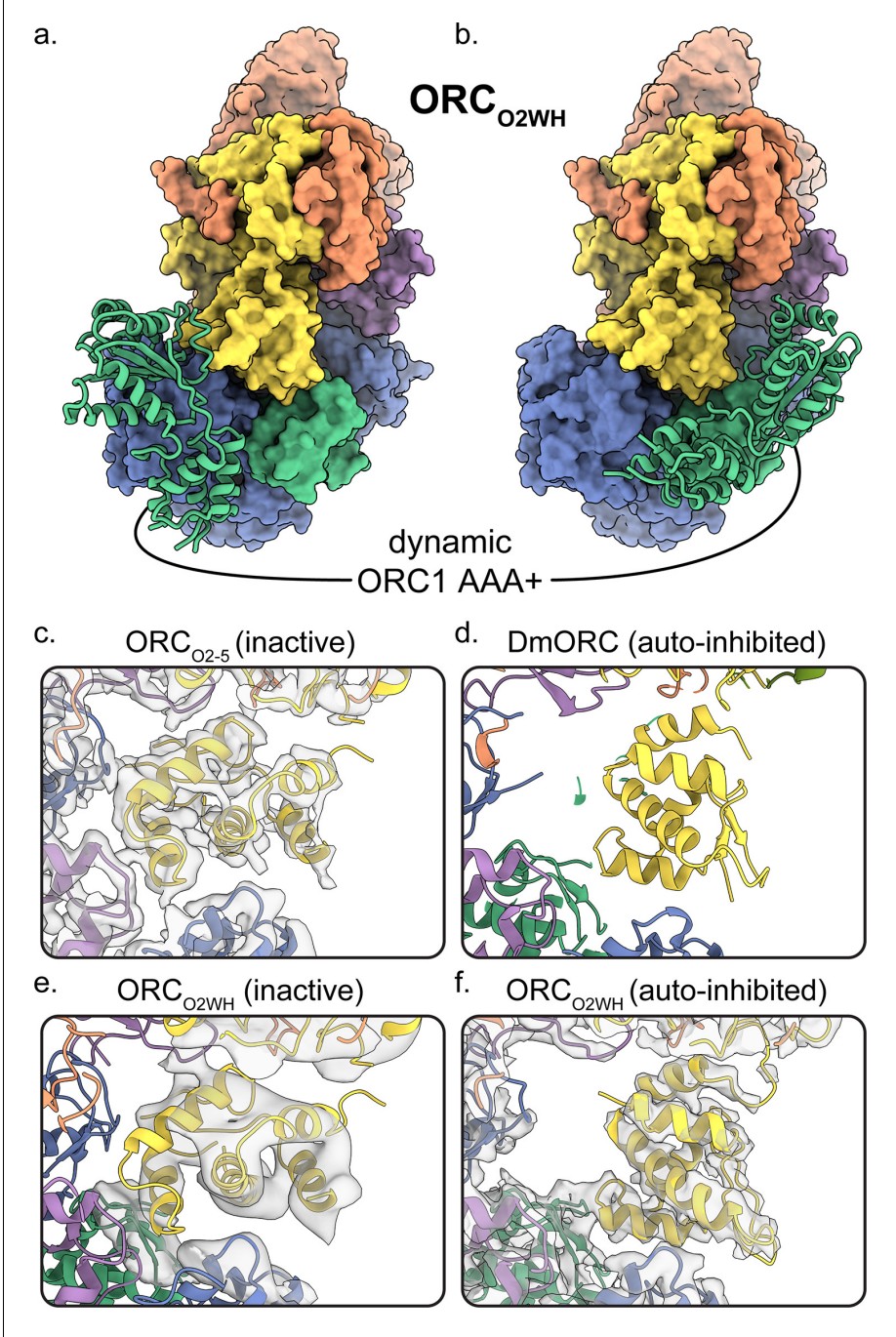

**Figure 2.** Dynamic states of ORC1 AAA+ and ORC2 WH domains. 3D classification ORC$_{O2WH}$ was performed with the density map low-pass filtered to 20 Å and an E-step resolution limit of 20 Å. Two 3D classes from ORC$_{O2WH}$ contained densities for ORC1 AAA+ domain. The ORC1 AAA+ domain was fit into these densities which revealed the domain approaching the ORC1·4 position (**a**) and the auto-inhibited position (**b**). (**c-f**) In the ORC core, the ORC2 WHD was observed in two conformations: an alternate state, called the inactive state (**c, e**), and the auto-inhibited state (**f**), similar to the DmORC2 WHD (PDB ID: 4xgc) (**d**). The ORC$_{O2WH}$ inactive state (**e**) was determined by focused classification of the ORC$_{O2WH}$ with a tight mask around ORC2 WH.

The online version of this article includes the following figure supplement(s) for figure 2:

**Figure supplement 1.** The dynamic states of ORC$_{O2WH}$ and ORC$_{O1AAA}$.

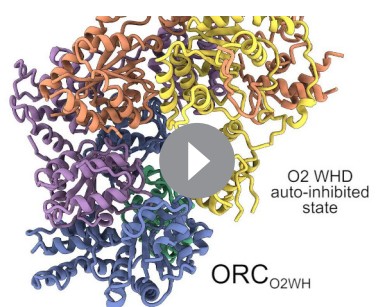

**Video 1.** The conformations of ORC2 WHD in ORC$_{O2-5}$ and ORC$_{O2WH}$. The ORC2 WHD in ORC$_{O2WH}$ is in a similar conformation to the autoinhibited state observed in DmORC (*Bleichert et al., 2015*). The ORC2 WHD must undergo a significant ~135°rotation to bind in the conformation observed in ORC$_{O2-5}$.
https://elifesciences.org/articles/58622#video1

appearance merits. Nonetheless, these maps provided valuable information about the architecture and dynamics of the complex (see below).

Three ATP nucleotides were well resolved in the ORC maps (*Figure 1g* and *Figure 1—figure supplement 1*), consistent with previous observations (*Tocilj et al., 2017*). Although not provided in the sample buffer, magnesium ions were also resolved in the maps and were likely retained from the protein expression lysate. Only in the ORC$_{O2WH}$ structure was an ATP absent at the ORC1·4 site due to the displacement of the ORC1 AAA+-domain (*Figure 1c*, *Figure 2a,b* and *Video 2*) and consequential disruption of the ATP binding site. The architectures of the ORC1·4 and ORC4·5 sites are characteristic of AAA+ ATPase proteins, where the Walker-A and Walker-B motifs of the RecA-fold of one subunit coordinate the ATP molecule on one face and three conserved basic residues, one from the same subunit and two from the other subunit coordinate the other face (*Figure 1g*). At the ORC1·4 interface, for example, these three basic residues are R720 of the ORC1 lid domain, acting as the trigger (or sensor-II), the ORC4 finger R209, and the ORC4 tether R205 coordinating the other face (*Figure 1g*). As mentioned, the only site that has been shown to have ATPase activity is the ORC1·4 site, and mutations at this site are associated with the developmental disorder Meier-Gorlin Syndrome (*Tocilj et al., 2017*). In contrast to ORC1·4 and ORC4·5, the ORC5·3 interface appears considerably more flexible and, in fact, ORC3 does not seem to participate in ATP coordination (*Figure 1g*). Nonetheless, we do observe density for an ORC3 region proximal to the ATP site in low-pass filtered maps which indicate dynamic movements of the unresolved residues 86–94.

The structures of ORC2, ORC3, and the WHD of ORC5 have significantly improved resolution compared to the previous published HsORC structure (*Tocilj et al., 2017*). ORC$_{O1AAA}$ was especially improved in this region due to the 3D focused refinement we applied (*Figure 1—figure supplement*

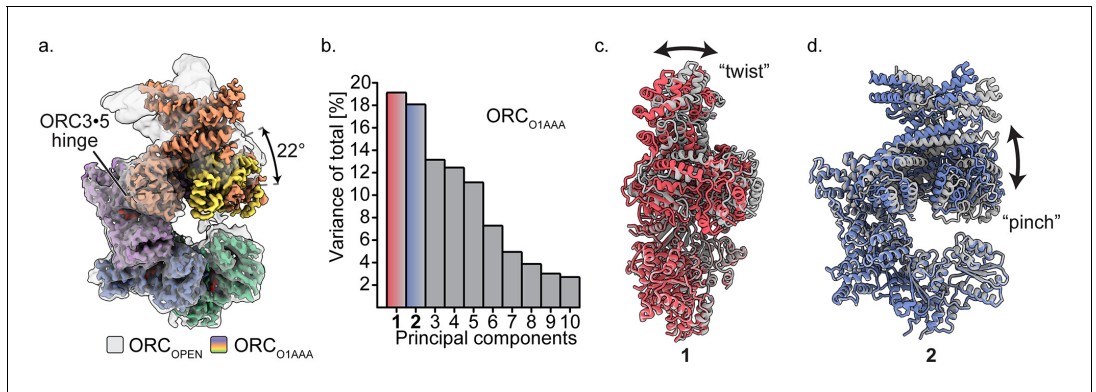

**Figure 3.** Multibody refinement of ORC$_{O1AAA}$. (a) Overlay of ORC$_{OPEN}$ and ORC$_{O1AAA}$ illustrates a large conformational change of ORC2 and ORC3. ORC2·3·(5 WH) and ORC1·4·(5 AAA+) were designated as two bodies for Relion's multibody refinement. (b) Principal component analysis revealed two prominent movements of ORC$_{O1AAA}$ highlighted in red and blue. (c) Component 1 undergoes twist movement of ORC2·3 across the ORC1 AAA+ domain. (d) Component 2 undergoes a pinch movement of ORC2·3 towards the ORC1 AAA+ domain. Multibody Refinement results of ORC$_{O2WH}$ and ORC$_{OPEN}$ are available in *Figure 3—figure supplement 1*. To validate the Multibody Refinement, Cryosparc's 3D Variability analysis was performed and yielded the same prominent movements. The movements from Multibody Refinement and 3D variability can be visualized in *Video 3*.
The online version of this article includes the following figure supplement(s) for figure 3:

**Figure supplement 1.** Relion's Multibody Refinement (*Nakane et al., 2018*) of ORC conformations.
**Figure supplement 2.** Extension of ORC3 N-terminal α-helix in ORC$_{OPEN}$.

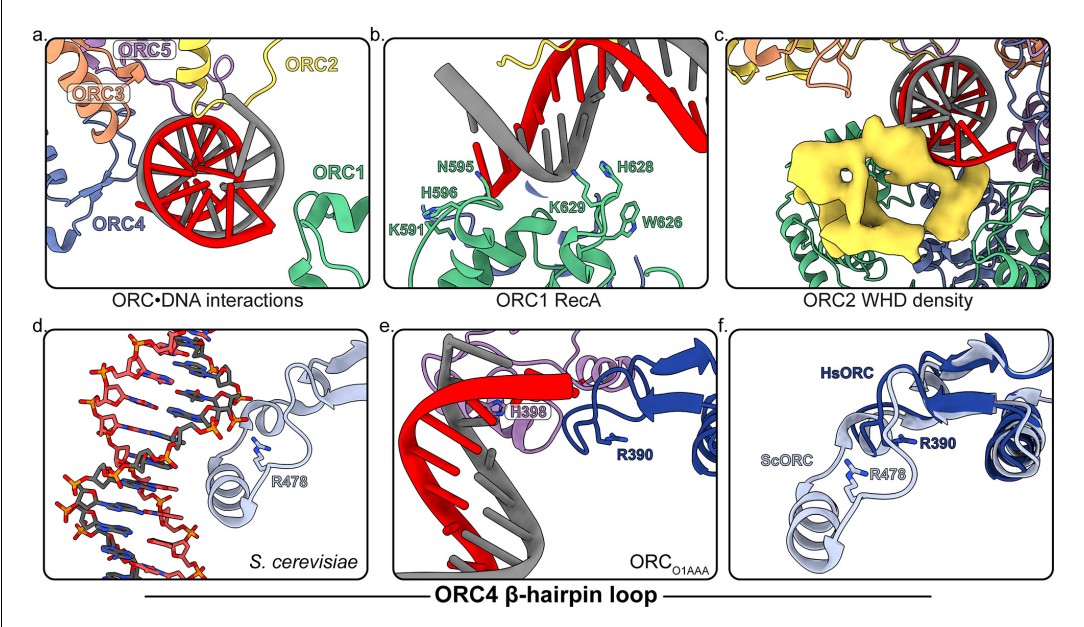

**Figure 4.** Several regions of ORC contact the endogenous DNA in ORC$_{DNA}$. (**a**) ORC subunits 1–5 contain regions with positively-charged residues proximal to the DNA. (**b**) The ORC1 RecA-fold contains several positively-charged residues in close proximity to DNA. (**c**) The ORC2 WHD, shown as a density map due to the lower resolution of this domain, moves to interact with the DNA in the core of the complex. (**c**) The β-hairpin loops of ORC4 and ORC5 contain positively-charged residues proximal to the DNA. The DNA depicted consists of non-specific sequence. (**d**) The ORC4 β-hairpin loop of *S. cerevisiae* contains an α-helix that inserts into the major groove of DNA making sequence-specific interactions with DNA bases. (**e**) The β-hairpin loop in ORC$_{DNA}$ does not contain an α-helix. However, ORC4 R390 in humans may interact with the DNA backbone similar to ORC4 R478 in *S. cerevisiae*. (**f**) The superimposition of the ScORC and HsORC β-hairpin loops illustrates the significant difference between the regions.

The online version of this article includes the following figure supplement(s) for figure 4:

**Figure supplement 1.** DNA density observed in the ORC core.
**Figure supplement 2.** Endogenous Sf9 DNA co-purifies with ORC1-5.
**Figure supplement 3.** The DNA in the core of ORC$_{DNA}$ adopts multiple conformations.
**Figure supplement 4.** ORC1 sequence alignment with a focus on the region in close proximity to DNA in the ORC$_{DNA}$ structure.
**Figure supplement 5.** ORC4 sequence alignment with a focus on the β-hairpin loop region near DNA.

*7*). The close intertwine between ORC2 and ORC3 in ScORC and DmORC structures is also apparent in the structures of HsORC presented here (*Figure 1—figure supplement 8*). The N-terminal regions of ORC2 and ORC3 wrap around each other contributing to their tight binding. In fact, ORC2 and ORC3 localize to centromeres in mitosis and may function independently of the other ORC subunits (*Craig et al., 2003*; *Prasanth et al., 2004*; *Prasanth et al., 2010*). The solvent-accessible surface area buried between ORC2 and ORC3 (3296 Å$^2$) is significantly larger than the next closest interacting subunits ORC4 and ORC5 (2707 Å$^2$). In the ScORC structure, the ORC2 N-terminal region wraps around the WHD of ORC3 and forms a helix on the opposite end (*Li et al., 2018*). In the HsORC structures, the ORC2 N-terminal helix (aa239-249) is present in the ORC$_{O1AAA}$ and ORC$_{O2WH}$ maps, while the loop (aa250-267) wrapping around ORC3 is dynamic and thus unresolved (*Figure 1—figure supplement 8*). In addition, the ORC3 N-terminal region (residues 10–14) loops around the RecA-like fold of ORC2 and forms a β-strand that adheres to one of the ORC2 RecA-like β-strands (aa335-339).

## Dynamic states of ORC1 AAA+ and ORC2 WH

The ORC$_{O1AAA}$ and ORC$_{O2WH}$ structures illustrate the dynamic movements of ORC1 AAA+ and ORC2 WH domains relative to each other (*Figure 1c,d*). Other than the ORC1 AAA+ and ORC2 WHD, the architectures of the two conformations are similar. In both maps, density for the missing domains can be seen in 3D classifications with a 20 Å low-pass filtered template and E-step resolution limit of 20 Å (*Figure 2a,b*, *Figure 2—figure supplement 1* and *Video 2*). In the ORC$_{O1AAA}$

structure, the ORC2 WHD undergoes conformational changes but does not change its overall position in the complex (*Figure 2—figure supplement 1a*). In contrast, in the $ORC_{O2WH}$ structure, the ORC1 AAA+ rotates ~90° from a position proximal to the ORC4 AAA+ domain where it completes the major ATP binding site in the $ORC_{O1AAA}$ structure, to a position proximal to the ORC1 and ORC5 WHDs (*Figure 2a,b*, *Figure 2—figure supplement 1* and *Video 2*), disrupting the ORC1·4 ATP binding site. The latter position resembles the auto-inhibited state observed in the DmORC structure (*Bleichert et al., 2015*). This state was also observed in 2D classifications of the human ORC (*Bleichert et al., 2018*). While ORC1 AAA+ is locked in the auto-inhibited state in DmORC, the ORC1 AAA+ domain in human ORC appears to be dynamic along a continuum of positions between two extreme states: the auto-inhibited state observed in the DmORC and when it forms the ORC1·4 interface with ORC4 as observed in $ORC_{O1AAA}$. As mentioned, the ATPase site between the ORC1 and ORC4 is the only site that exhibits activity and mutations at the site are associated with the developmental disorder Meier-Gorlin Syndrome, but the functional role of ATP hydrolysis is unknown. The ORC1 AAA+ domain's dynamic movements may be a consequence of ORC1·4 ATP hydrolysis, but further investigation is necessary. On the other hand, the ORC4·5 and ORC5·3 subunit interfaces are stable, and do not appear to be disrupted, which may be a consequence of their inability to hydrolyze ATP. Based on this idea and previous studies using mutant ORC4 and ORC5 subunits (*Siddiqui and Stillman, 2007*), we conclude that ATP promotes the assembly and stability of the human ORC, which, unlike yeast ORC, is assembled and disassembled throughout the cell division cycle. Overall, the $ORC_{O1AAA}$ and $ORC_{O2WH}$ populations reveal a dynamic switch between the ORC1 AAA+ and ORC2 WHD where the locking into position of one domain within the ORC ring-shaped core kicks the other domain into a dynamic state. These domains are accommodated simultaneously only in the $ORC_{OPEN}$ and $ORC_{DNA}$ complexes, two conformations that separate the ORC1 AAA+ and ORC2 WH domains by widening the gap between ORC1 and ORC2 and by mutual electrostatic attraction to DNA, respectively (see below).

A minor ORC2 WHD conformation was observed in further 3D classifications of the $ORC_{O2WH}$ particles (*Figure 2e*). Secondary structure elements can be seen in the ORC2 WHD region that are clearly different from the auto-inhibited conformation. In fact, this conformation resembles the completely collapsed ORC2 WHD conformations in the $ORC_{O2-5}$ structure where the domain is further tucked into the core of the complex compared to the auto-inhibited state (*Figure 2c*). Although the density region of the ORC2 WHD alternative state in $ORC_{O2WH}$ is of low resolution, we were confidently able to fit this region once we modelled the ORC2 WHD inactive state in the well-resolved $ORC_{O2-5}$ structure (*Figure 2c*). Notably, this alternative conformation does not resemble any of the conformations of ORC2 WHD in any previously reported structures of the various ORCs (*Bleichert et al., 2015*; *Bleichert et al., 2018*; *Li et al., 2018*; *Tocilj et al., 2017*; *Yuan et al., 2017*). The buried ORC2 WHD appears to stabilize ORC2-5 complex prior to ORC1 binding, hence we termed this structure of the ORC2 WHD the inactive state (*Figure 1b* and *Figure 2c,e*). The binding of ORC1 seems to facilitate the release of the O2 WHD from this tucked-in state, generating a more labile structure for downstream events. In both the yeast OCCM and ORC-DNA structures (*Li et al., 2018*; *Yuan et al., 2017*), the ORC2 WHD interacts with the origin DNA and is far away from the gap between the ORC1 and ORC2 AAA+ domains, forming a collar of ORC WHDs that surround the DNA. Thus the movement of the ORC2 WHD seems to be a universal feature of controlling DNA access to the ring-shaped core of ORC and hence binding to DNA.

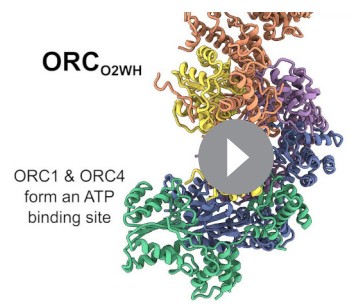

$ORC_{O2WH}$

ORC1 & ORC4 form an ATP binding site

**Video 2.** The dynamic conformations of ORC1 AAA+ in $ORC_{O2WH}$. A low-pass filtered 3D classification of $ORC_{O2WH}$ revealed an ORC1 AAA+ position near the ORC1·4 interface that forms the ATP binding site and an ORC1 AAA+ position near the auto-inhibited state disrupting the ORC1·4 ATP binding site.
https://elifesciences.org/articles/58622#video2

## Hinge movement arising from the ORC3·5 interface

The $ORC_{OPEN}$ structure displays a conformation similar to what was reported previously for HsORC and ScORC (*Figure 1e*). When aligned to $ORC_{O1AAA}$, a hinge in $ORC_{OPEN}$ is evident and

originates from the somewhat loose ORC3·5 interface, allowing the ORC2·3 subunits (including the ORC5 WH) to rotate (22° rotation angle) away from the ORC1·4·5 motor module in a hinge motion (*Figure 3a*). In fact, these two portions of ORC can exist as two separate stable complexes, as evident by the isolation of these stable sub-complexes previously (*Tocilj et al., 2017*). In addition, the ORC3 N-terminal region of ORC$_{OPEN}$ consists of an extended, straightened α-helix (aa42-86) that is bent in the other ORC conformations by 32° (*Figure 1e*, *Figure 3—figure supplement 2*). This long helix in ORC$_{OPEN}$ may provide structural stability to the open conformation. The ORC3 straight helix is also present in the ScORC structure (*Li et al., 2018*; *Yuan et al., 2017*), while the helix is bent in the DmORC structure (*Bleichert et al., 2015*). Interestingly, both the ORC1 AAA+ and O2 WHD are ordered in the open conformation. The extra space in the ORC ring opening reduces clashes between the ORC2 WHD and ORC1 AAA+ domain and allows both domains to adopt a fixed orientation relative to their mutually-exclusive static positioning observed in the more closed ORC$_{O1AAA}$ and ORC$_{O2WH}$ structures (*Figure 2a,b*, *Figure 2—figure supplement 1*).

To assess the prominent movements around the ORC3·five interface, we performed multibody refinement in Relion on the ORC$_{O1AAA}$, ORC$_{O2WH}$, and ORC$_{OPEN}$ particle populations (*Figure 3b–d* and *Video 3*). Subunits ORC2·3 (including ORC5 WHD) and ORC1·4·5 of HsORC were designated as two separate masked bodies in Relion's multibody refinement to determine their independent movements (*Nakane et al., 2018*) (ORC1 was isolated as a third masked body in multibody refinement, but the experiment did not provide conclusive results, most likely due to the relatively small size of ORC1). The components of the masked bodies and ORC movements were further validated by performing Cryosparc's unbiased 3D Variability Analysis on the ORC$_{O1AAA}$ particles (*Video 3*; *Punjani and Fleet, 2020*). Compared to Multibody Refinement, 3D Variability Analysis does not rely on user input to determine regions that move independently of one another (*Punjani and Fleet, 2020*). Both Multibody Refinement and 3D Variability Analysis revealed the same prominent movements in the ORC$_{O1AAA}$ particle population. For all three ORC populations, multibody refinement generated two components (specific movements of ORC) that contributed significantly more to the overall variance (total movement of ORC) (*Figure 3—figure supplement 1b* and *Video 3*). These movements contributed between 17% to 23% of the total variance of each particle population. The two movements can be described as twisting and pinching, respectively, both emanating from the hinge at the ORC3·5 interface (*Figure 3c,d* and *Video 3*). The ORC twisting motion involves ORC1 and ORC2 sliding over each other, while the pinching motion involves ORC1 and ORC2 narrowing/widening the ORC ring opening between them (*Figure 3c,d* and *Video 3*). The twisting and pinching movements provide potential flexibility to ORC during replication initiation events, including DNA binding, movement along DNA, and/or complex release after MCM double hexamer formation.

In the global refinement of ORC, the ORC2·3 regions lining the gap with ORC1 suffered the most in terms of alignment. In refinements, the center of mass for alignments is located on ORC1·4·5 and, since ORC2·3 moves independently of ORC1·4·5, the edges of ORC2 and ORC3 were misaligned. However, the multibody/focused refinement significantly improved the ORC2·3 region of the map (*Figure 1—figure supplement 7*). The N-terminus of ORC3, which wraps around ORC2 and extends the β-sheet of

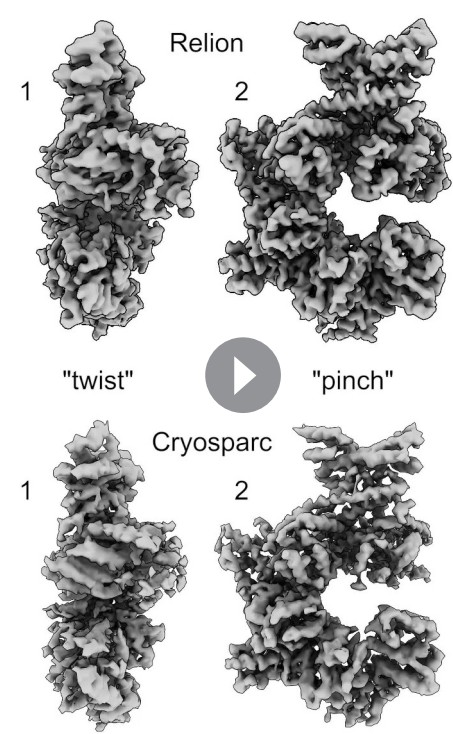

**Video 3.** The ORC3·5 interface acts as a flexible hinge. Relion's Multibody Refinement and Cryosparc's 3D Variability Analysis revealed the same two prominent movements which resemble a twist and pinch motion, respectively.
https://elifesciences.org/articles/58622#video3

the RecA-like fold of ORC2 with an additional strand, was clearly resolved in the focused-refined map. In addition, the density of an ORC2 loop region (residues 361–368), which comes into close proximity with bound DNA in the ORC$_{DNA}$ map (see below), is significantly improved. The improved density maps of ORC further validates the multibody refinement parameters and, therefore, the flexibility at the ORC3·5 interface.

## DNA bound to HsORC

One class from the ORC1-5 sample contained clear density for DNA in the core of the complex and yielded a 4.3 Å map into which DNA could be fit into the ORC1-5 structure (ORC$_{DNA}$) (*Figure 1f*, *Figure 4*, *Figure 4—figure supplement 1*). DNA was detected in the affinity-purified, recombinant HsORC expressed in Sf9 insect cells and the presence of this DNA species was only detected when ORC was expressed (*Figure 4—figure supplement 2a*). Furthermore, a post 2D classification of the DNA-bound HsORC yielded classes with strong density in the core of the complex (*Figure 4—figure supplement 2b*). The bright appearance of this density in the 2D classification likely represents the heavy phosphorous atoms in the DNA backbone since heavy atoms scatter electrons to a greater extent and to higher angles in TEM (*Kellenberger et al., 1986*). Since DNA-binding of HsORC is not sequence specific, the DNA is likely to have multiple orientations along its axis in the core, consistent with further 3D classification of the DNA-bound ORC populations (*Figure 4—figure supplement 3*). In other words, the DNA fragment visualized is a result of a superposition of sequences along the DNA fragment. The overall conformation of ORC$_{DNA}$ resembles the more closed conformations of ORC$_{O1AAA}$ and ORC$_{O2WH}$, albeit with ORC2 WHD in a different conformation (see below). The structure shows DNA contacts by all five subunits of ORC, with the most significant interactions occurring with the ORC1 RecA-fold and ORC2 WHD (*Figure 4a–c*). In the ORC1 RecA-fold, two loops (residues 593–596 and 626–630, respectively) at the N-terminal regions of two α-helices are in close contact to the DNA. Importantly, there are several positively-charged and aromatic residues in the loops (K591, N595, H596 W626, H628, K629) that appear to be involved in binding the negatively-charged DNA backbone (*Figure 4b*). Sequence alignments show that while most of these residues are conserved in vertebrate species, they differ in insects, plants and fungi (*Figure 4—figure supplement 4*). The ORC2 WHD is also in close contact with the DNA and this domain undergoes significant secondary structure rearrangement, which prevented precise fitting of modelled ORC2 WHD into the region, and the resolution is not high enough for unambiguous model building of this region (*Figure 4c*). The ORC2 WHD may adopt multiple conformations in this structure similar to the multiple ORC2 WHD states observed in DNA-bound ScORC (*Li et al., 2018*). Nevertheless, clear secondary structure is visible in the ORC2 WHD region of the map, in which, a long helix has significant contact with the DNA density (*Figure 4c*). In addition, the initiation specific motif (ISM) of ORC2 (residues 364–368) in the RecA-like fold is near the DNA substrate. Specifically, the positively-charged ORC2 residue R367 emanating from this loop, appears to mediate the interaction with DNA. The ORC2 ISM in humans may play an important role in binding DNA similar to the ORC2 ISM in ScORC where W396 is buried in the minor groove of DNA (*Li et al., 2018*; *Yuan et al., 2017*). The β-hairpin loop in ORC5 of HsORC also consists of a positively-charged residue H398 in close proximity to DNA (*Figure 4a,f*). Similarly, the β-hairpin loop in ORC5 of ScORC contains positively-charged residues (N438, K439, N440, K447, R449) extending into the major groove of DNA (*Li et al., 2018*).

The Orc4 WHD plays an important role in origin-specific binding in *S. cerevisiae* (*Hu et al., 2020*; *Li et al., 2018*; *Yuan et al., 2017*). In *S. cerevisiae*, this domain contains an α-helix insertion in the β-hairpin wing (residues 481–489) that inserts into the major groove of the ARS-sequence DNA and is a key contributor to conferring sequence specificity for origin recognition (*Figure 4d*). The homologous ORC4 region in HsORC is visible in the ORC$_{O1AAA}$ cryoEM density map allowing us to now model this region (*Figure 4e*). This region in HsORC is 20 residues shorter than the ScORC segment and does not contain an α-helix (*Figure 4f* and *Figure 4—figure supplement 5*; *Hu et al., 2020*). The missing helix in HsORC likely reduces its ability to bind specifically to a DNA sequence resulting in the requirement of other mechanisms to establish stable origin interactions. This loss of specificity also contributes to the multiple conformations of DNA observed in the core of human ORC (*Figure 4—figure supplement 3*). Despite the missing helix, the shorter ORC4 loop contacts the DNA in the ORC$_{DNA}$ structure proximal to R390 (*Figure 4e*), a positively-charged residue that may participate in DNA-backbone interactions similar to R478 in ScORC4 (*Figure 4d*). Based on sequence alignments (*Figure 4—figure supplement 5*), the ORC4 helix insertion has likely evolved in a small

clade of *Saccharomyces*-related budding yeast species in tandem with a loss of RNA interference (RNAi)-mediated transcriptional gene silencing, while the region is absent in most other eukaryotic organisms, including fungi, plants, and animals, which use RNAi for silencing (*Hu et al., 2020*). Since HsORC does not display DNA sequence specific DNA binding (*Vashee et al., 2003*; *Vashee et al., 2001*), the significant difference in ORC4 between budding yeast and human ORC contributes to the sequence-dependent and -independent origin binding in these species (*Figure 4f*). HsORC binds to specific locations in chromosomes (*Miotto et al., 2016*) and unlike ScORC, it is most likely targeted to these sites by binding to specific histones and their modifications (*Hossain and Stillman, 2016*; *Kuo et al., 2012*; *Long et al., 2020*).

## Concluding remarks

DNA replication initiation is a highly dynamic, multi-complex process that requires essential coordination by ORC. Our findings uncover several dynamic conformations of HsORC which have implications in facilitating initiation events including DNA engagement, CDC6 recruitment, and MCM complex assembly onto DNA. We determined four structures of ORC that demonstrate the previously unappreciated flexibility of the complex. The $ORC_{O1AAA}$ and $ORC_{O2WH}$ consists of two dynamic domains, ORC2 WH and ORC1 AAA+, respectively, that repel each other from adopting their fixed positions. The domain dynamics are possibly linked to the catalytic activity at the ORC1·4 ATPase site, warranting further analysis to determine the molecular basis and the biological role of these domain movements. While the $ORC_{OPEN}$ structure largely resembles the ScORC conformation in the budding yeast OCCM complex and the previously reported HsORC structure, it unexpectedly reveals a significant hinge point at the ORC3-5 interface that generates a twist and pinch motion. The twist motion may allow ORC to move on DNA with a scanning mechanism to establish origin sites and the pinch motion may be involved in facilitating ORC to accommodate DNA into its central cavity via the partial ring opening and the subsequent incorporation of CDC6 into the ORC-DNA complex. The ATPase activity of ORC may promote the dynamic movement of the ORC1 and ORC2 subunits during assembly or prior to origin recognition. Lastly, the $ORC_{DNA}$ structure contains endogenous DNA bound in the core of ORC and reveals regions in the core that are in close proximity to the bound DNA. These observations along with previous studies on the dynamic, ATP-dependent assembly of ORC (*Siddiqui and Stillman, 2007*) suggest a model in which a pre-assembled ORC2-5 complex binds to ORC1 and promotes the opening of the ORC core to facilitate DNA binding. Since ORC1 appears to be the first ORC subunit to bind to chromosomes during mitosis (*Kara et al., 2015*; *Okuno et al., 2001*), ORC1 may recruit the other subunits via the pathway outlined in *Figure 5*. These interactions between ORC1 and ORC2-5 may be facilitated by intrinsically disordered regions (IDRs) the predicted N-terminal regions of ORC1 and ORC2 forming condensates via phase transitions (*Parker et al., 2019*). Proteins containing IDRs are difficult to express and by their nature contain large unstructured regions. We therefore removed the ORC1 N-terminal region from our construct and we were not able to resolve the ORC2 N-terminal region. Due to the absence of these regions, IDRs appear unnecessary for ORC assembly. Finally, it is known that the C-terminus of yeast Mcm3 binds to and activates the ATPase activity of the ORC-Cdc6 complex on DNA, possibly preventing futile loading of incomplete Mcm2-7 hexamers (*Frigola et al., 2013*). It is also possible that if the C-terminus of Mcm3 stimulates the ATPase activity of ORC, it may induce movement of the ORC1 AAA+ and a switch from the active to the inactive state of ORC after complete pre-RC assembly. Comparing the human and yeast ORC DNA contacts provides insights into the different sequence specificity requirements of ORC-DNA binding between the species. Our study establishes the foundation for further exploration of the critical determinants in metazoan replication origin establishment.

# Materials and methods

## Protein preparation

Codon optimized human Origin Recognition Complex (HsORC) synthetic genes [NP_004144.2 HsORC subunit 1 (HsORC1), NP_006181.1 HsORC subunit 2 (HsORC2), NP_862820.1 HsORC subunit 3 (HsORC3), NP_859525.1 HsORC subunit 4 (HsORC4), NP_002544.1 HsORC subunit 5 (HsORC5)] were cloned into the MultiBac baculovirus expression system (*Bieniossek et al., 2008*).

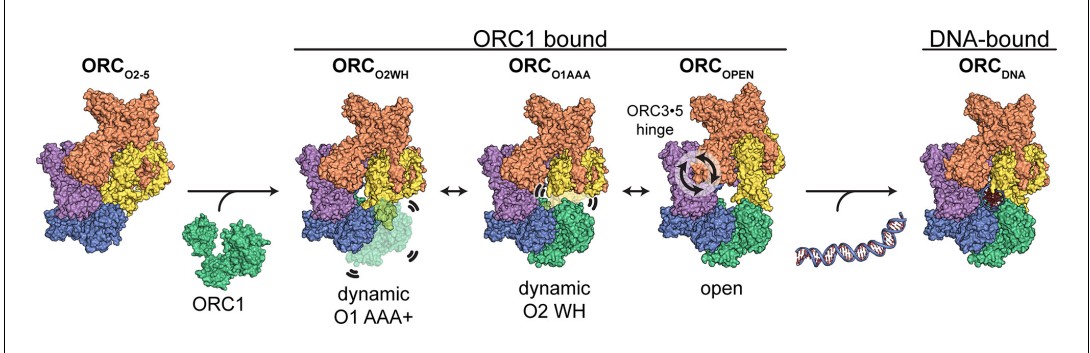

**Figure 5.** Proposed model for ORC dynamics. The binding of ORC1 to ORC2-5 opens the complex and facilitates dynamic movements of ORC1 AAA+ and ORC2 WH at the ring opening and a hinge motion at the ORC3·5 interface. The binding of DNA facilitates positioning of ORC1 AAA+ and ORC2 WHD on DNA for interaction to establish replication origin recognition.

The ORC1 subunit was truncated at the N-terminus to facilitate stability and monodispersity of human ORC based on the previous study (*Tocilj et al., 2017*), while the remaining subunits were expressed as full-length proteins.

For recombinant expression of the hetero-pentameric HsORC1-5, gene-coding sequences were cloned into the pFL vector (pH promoter HsORC1 residues 471–861), pUCDM vector (pH promoter HsORC4 residues 1–436 (full-length), p10 promoter HsORC3 residues 1–712 (full-length) and pSPL vector (pH promoter HsORC2 residues 1–577 (full-length), p10 promoter HsORC5 residues 1–435 (full-length)). A StrepTag was introduced to the N-terminus of ORC3 for affinity purification. Sf9 insect cells were infected with baculovirus in Hyclone CCM3 media (GE Healthcare Life Sciences, Pittsburgh, PA) for 48 hr. For recombinant expression of HsORC2-5, the baculovirus system was identical to that of HsORC1-5, except that the HsORC1 was full-length (residues 1–861). However, after affinity purification of HsORC2-5, ORC1 was not present in the protein sample. The purification steps of HsORC1-5 and HsORC2-5 were identical.

All purification steps were performed at 4˚C. Cell pellets were thawed on ice and resuspended in lysis buffer (50 mM HEPES-NaOH (pH 7.5), 200 mM KCl, 6.5 mM dithiothreitol (DTT), 1.7 mM adenosine triphosphate (ATP)). Following sonication, the lysate was centrifuged for 45 min at 143,000 g and the supernatant was loaded onto 5 ml of StrepTactin Super-flow resin (IBA, Goettingen, Germany). The beads were washed and recombinant HsORC1-5 complex was eluted with lysis buffer, which contained 5 mM desthiobiotin. The elution was concentrated and loaded onto a Superdex 200 Increase 10/300 GL size exclusion column (GE Healthcare Life Sciences, Pittsburgh, PA) equilibrated with minimal buffer (25 mM HEPES-NaOH (pH 7.5), 80 mM KCl, 3.3 mM dithiothreitol (DTT), 0.8 mM adenosine triphosphate (ATP)).

## Cryo-electron microscopy sample preparation

CryoEM sample preparations for HsORC1-5 and HsORC2-5 were identical. ORC samples were diluted to 0.8 mg/ml and Lauryl Maltose Neopentyl Glycol (LMNG, Anatrace, Maumee, OH) were added to 0.05% w/v to reduce preferred orientations and promote thin layers of ice over grid holes. For cryoEM grid preparations, we applied 4 µl of ORC at a final concentration of 0.8 mg/ml to a glow-discharged Quantifoil R 2/1 300 mesh copper grid, incubated for 10 s at 25˚C and 90% humidity, blotted for 2.9 s, and then plunged the grid into liquid ethane using a Leica Automatic Plunge Freezer EM GP2 (Buffalo Grove, IL).

## Cryo-electron microscopy data acquisition

Cryo-EM data were collected on a FEI/ThermoFisher Titan Krios transmission electron microscope (TEM) operating at 300 keV. Dose-fractionated movies were collected using a Gatan K2 Summit direct electron detector operating in electron counting mode. For HsORC1-5, movies were collected with 30 frames during a 6 s exposure at an exposure rate of 2.2 e$^-$/A$^2$/frame, resulting in a cumulative exposure of 66 e$^-$/Å$^2$. For HsORC2-5, 35-frame movies were collected during a 7 s exposure at an exposure rate of 2.2 e$^-$/Å$^2$/frame, resulting in a cumulative exposure of 77 e$^-$/Å$^2$. The EPU data

collection software (ThermoFisher, Hillsboro, OR) was used to collect 9068 micrographs for HsORC1-5 and 5549 for HsORC2-5 micrographs by moving to hole centers and image shifting to five positions within the hole to acquire five exposures at 130,000x nominal magnification (1.07 Å/pixel at the specimen level), with a nominal defocus range of −1.0 to −2.4 μm. Data collection information is summarized in *Supplementary file 1*.

## Image processing and 3D reconstruction

WARP was used to perform real-time image pre-processing (motion correction, CTF estimation, particle picking) during cryo-EM data collection for both HsORC1-5 and HsORC2-5 (*Tegunov and Cramer, 2019*). Particle picking was performed with the BoxNet pretrained neural network bundle employed in WARP and implemented in TensorFlow. A particle diameter of 160 Å and a threshold score of 0.6 yielded 2,112,391 particle coordinates for HsORC1-5 and 494,812 particle coordinates for HsORC2-5. These particles were initially subjected to a 2D classification in Cryosparc v2 (*Punjani et al., 2017*). A subset of particles were selected for *ab initio* reconstruction in Cryosparc v2 to generate an initial model for 3D classification in Relion 3.0 (*Figure 1—figure supplements 1–2*; *Zivanov et al., 2018*). Relion 3.0 was used for all remaining processing steps, including reprocessing of the micrographs (Relion's implementation of motion correction), CTF estimation (Gctf) (*Zhang, 2016*; *Zivanov et al., 2018*).

For HsORC2-5, a total of 494,812 particles (from WARP/BoxNet coordinates) were initially extracted from micrographs using an unbinned box size of 320 pixels binned to a box size of 80 pixels (4 x bin) for 3D classification in Relion. The particles were classified into five classes using the scaled and low-pass filtered *ab initio* reconstruction from Cryosparc v2 as a reference map and a regularization parameter of four. The 209,479 particles contributing to the best 3D class average containing well-defined structural features were further unbinned, classified, and refined to yield a 3.7 Å map of 168,444 particles. To further improve the resolution, we performed Bayesian polishing and CTF refinement, twice each, to re-refine the particle stacks, which yielded a map with a reported resolution of 3.3 Å. The ORC2 WHD in the refined map was significantly unresolved, therefore, a 3D classification (3 classes, no alignment, regularization parameter of 100) with a mask encompassing the ORC2 WHD was performed that yielded a map with more complete density in the region. The 53,009 particles from the classification were further refined to yield a final map of 3.5 Å ($ORC_{O2-5}$).

For HsORC1-5, a total of 2,112,391 particles (from WARP/BoxNet coordinates) were initially extracted from micrographs using an unbinned box size of 320 pixels binned to a box size of 80 pixels (4 x bin) for 3D classification in Relion. The particles were classified into five classes using the scaled and low-pass filtered *ab initio* reconstruction from Cryosparc v2 as a reference map and a regularization parameter of four. The 855,373 particles contributing to the best 3D class average containing well-defined structural features were further unbinned, classified, and refined to yield four prominent density maps at resolutions of ~3.8 Å ($ORC_{O1AAA}$),~4.0 Å ($ORC_{O2WH}$),~4.0 Å ($ORC_{open}$), and ~4.3 Å ($ORC_{DNA}$). The observable structural features of the $ORC_{open}$ and $ORC_{DNA}$ maps were not consistent with the reported resolutions and appeared to suffer from issues with flexibility and misalignment, and preferred orientation and were, therefore, not further refined. In addition, the program 3DFSC (*Tan et al., 2017*) revealed that these maps also suffered from preferred orientation (*Figure 1—figure supplement 6*). The $ORC_{O1AAA}$ and $ORC_{O2WH}$ maps contained well-defined features consistent with their respective resolutions, but were not sufficiently resolved for atomic modeling. To further improve the resolution, we performed CTF refinement and Bayesian polishing to re-refine the particle stacks, which yielded reconstructed maps with a reported resolution of 3.2 Å ($ORC_{O1AAA}$) and 3.7 Å ($ORC_{O2WH}$). Several regions of the $ORC_{O1AAA}$ map (particular the regions of ORC2 and ORC3 that line the gap in the ORC open-ring structure) did not contain structural features that are consistent with 3.2 Å cryo-EM density. Considering several 3D classes consisted of ORC2 and ORC3 in different orientations relative to the rest of the complex, we performed multibody refinement (*Nakane et al., 2018*) using two masked regions: ORC2·3·5 WHD and ORC1·4·5 AAA+. The chosen locations of the masks were further validated by performing Cryosparc's unbiased 3D Variability Analysis (*Punjani and Fleet, 2020*) on the $ORC_{O1AAA}$ particle population which revealed the same movements as the Multibody Refinement. The multibody refinement improved the resolution and map quality of the region of ORC2·3 mentioned above (*Figure 3—figure supplement 2*). A final composite $ORC_{O1AAA}$ map of the focused and non-focused refinements was generated for atomic model building and refinement using the 'vop max' operation in UCSF

Chimera (*Goddard et al., 2007*). Final cryoEM density map information is summarized in *Supplementary file 1*.

## Model building

The atomic model of HsORC (PDB ID: 5ujm) (*Tocilj et al., 2017*) was used as a template for model building in ORC$_{O1AAA}$ and map. Density in the ORC2 and ORC3 region of the previous structure were of low resolution and relied significantly on the *Drosophila melanogaster* ORC2 and ORC3 model as a template (*Bleichert et al., 2015*). In this work, the high-resolution map of ORC$_{O1AAA}$ allowed for precise modelling of each residue in Coot which resulted in regions that were rebuilt, and regions built for the first time (especially in ORC2 and ORC3). Subunits of the ORC$_{O1AAA}$ model were used as templates for model building into the ORC$_{O2-5}$, ORC$_{O2WH}$, ORC$_{OPEN}$, and ORC$_{DNA}$ maps. The ORC2 WHD of the ORC$_{O2WH}$ map was modelled initially using the ORC2 WHD from the DmORC atomic model (*Bleichert et al., 2015*). Each subunit of the models was rigid body fit using the 'fit in map' function in UCSF Chimera (*Goddard et al., 2007*). The resulting model, including nucleotides and coordinating metal ions, was subject to further model building using the COOT software package (*Emsley and Cowtan, 2004*), and then real-space refined with the PHENIX package (*Afonine et al., 2012*). The rotation angle calculations at the O3·5 hinge was determined using the 'align' function in UCSF ChimeraX (*Goddard et al., 2018*). The ORC$_{O1AAA}$ and ORC$_{OPEN}$ structures were aligned at ORC4, followed by alignment of ORC2·3 which generates a report containing the rotation angle between the aligned ORC2·3 subunits. UCSF ChimeraX (*Goddard et al., 2018*) was also used to generate the figures and buried surface areas were determined using the 'measure buriedarea' function in ChimeraX). Model statistics are summarized in *Supplementary file 1*.

## Acknowledgements

We thank members of the Joshua-Tor laboratory for helpful comments and suggestions. We thank the CSHL Mass Spectrometry Shared Resource, which is supported by Cancer Center Support Grant 5P30CA045508. This work was supported by a Ruth L Kirschstein National Research Service Awards National Institutes of Health (NIH) fellowship F32GM129923 (to MJJ) and by NIH grant GM45436 (to BS). LJ is an investigator of the Howard Hughes Medical Institute.

## Additional information

### Competing interests

Bruce Stillman: Reviewing editor, *eLife*. The other authors declare that no competing interests exist.

### Funding

| Funder | Grant reference number | Author |
|---|---|---|
| Howard Hughes Medical Institute | | Leemor Joshua-Tor |
| National Institutes of Health | F32GM129923 | Matt J Jaremko |
| National Academies of Sciences, Engineering, and Medicine | GM45436 | Bruce Stillman |

The funders had no role in study design, data collection and interpretation, or the decision to submit the work for publication.

### Author contributions

Matt J Jaremko, Kin Fan On, Conceptualization, Data curation, Formal analysis, Validation, Investigation, Visualization, Methodology, Writing - original draft, Writing - review and editing; Dennis R Thomas, Data curation, Formal analysis; Bruce Stillman, Conceptualization, Investigation, Writing - review and editing; Leemor Joshua-Tor, Conceptualization, Formal analysis, Supervision, Funding

acquisition, Validation, Methodology, Writing - original draft, Project administration, Writing - review and editing

## Author ORCIDs
Matt J Jaremko https://orcid.org/0000-0002-0340-4452
Kin Fan On https://orcid.org/0000-0002-0949-732X
Bruce Stillman http://orcid.org/0000-0002-9453-4091
Leemor Joshua-Tor https://orcid.org/0000-0001-8185-8049

## Decision letter and Author response
Decision letter https://doi.org/10.7554/eLife.58622.sa1
Author response https://doi.org/10.7554/eLife.58622.sa2

# Additional files

## Supplementary files
- Supplementary file 1. CryoEM and model statistics of all ORC structures.
- Transparent reporting form

## Data availability
All coordinates and cryoEM maps have deposited in the PDB and EMDB: ORC-O1AAA: PDB code: 7JPO EMDB code: EMD-22417 ORC-O2WH: PDB code: 7JPP EMDB code: EMD-22418 ORC-O2-5: PDB code: 7JPQ EMDB code: EMD-22419 ORC-OPEN: PDB code: 7JPR EMDB code: EMD-22420 ORC-DNA: PDB code: 7JPS EMDB code: EMD-22421.

The following datasets were generated:

| Author(s) | Year | Dataset title | Dataset URL | Database and Identifier |
| --- | --- | --- | --- | --- |
| Jaremko MJ, On KF, Thomas DR, Stillman B, Joshua-Tor L | 2020 | ORC-O1AAA | https://www.rcsb.org/structure/7JPO | RCSB Protein Data Bank, 7JPO |
| Jaremko MJ, On KF, Thomas DR, Stillman B, Joshua-Tor L | 2020 | ORC-O1AAA | https://www.ebi.ac.uk/pdbe/entry/emdb/EMD-22417 | Electron Microscopy Data Bank, EMD-22417 |
| Jaremko MJ, On KF, Thomas DR, Stillman B, Joshua-Tor L | 2020 | ORC-O2WH | https://www.rcsb.org/structure/7JPP | RCSB Protein Data Bank, 7JPP |
| Jaremko MJ, On KF, Thomas DR, Stillman B, Joshua-Tor L | 2020 | ORC-O2WH | https://www.ebi.ac.uk/pdbe/entry/emdb/EMD-22418 | Electron Microscopy Data Bank, EMD-22418 |
| Jaremko MJ, On KF, Thomas DR, Stillman B, Joshua-Tor L | 2020 | ORC-O2-5 | https://www.rcsb.org/structure/7JPQ | RCSB Protein Data Bank, 7JPQ |
| Jaremko MJ, On KF, Thomas DR, Stillman B, Joshua-Tor L | 2020 | ORC-O2-5 | https://www.ebi.ac.uk/pdbe/entry/emdb/EMD-22419 | Electron Microscopy Data Bank, EMD-22419 |
| Jaremko MJ, On KF, Thomas DR, Stillman B, Joshua-Tor L | 2020 | ORC-OPEN | https://www.rcsb.org/structure/7JPR | RCSB Protein Data Bank, 7JPR |
| Jaremko MJ, On KF, Thomas DR, Stillman B, Joshua- | 2020 | ORC-OPEN | https://www.ebi.ac.uk/pdbe/entry/emdb/EMD-22420 | Electron Microscopy Data Bank, EMD-22420 |

| | | | | | |
|---|---|---|---|---|---|
| Tor L | | | | | |
| Jaremko MJ, On KF, Thomas DR, Stillman B, Joshua-Tor L | 2020 | ORC-DNA | https://www.rcsb.org/structure/7JPS | RCSB Protein Data Bank, 7JPS | |
| Jaremko MJ, On KF, Thomas DR, Stillman B, Joshua-Tor L | 2020 | ORC-DNA | https://www.ebi.ac.uk/pdbe/entry/emdb/EMD-22421 | Electron Microscopy Data Bank, EMD-22421 | |

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
