## [Decision Letter]

**Acceptance summary:**

The structures that you report of multiple states of the human ORC complex, an essential replication complex in free and DNA-bound states, is a major advance. The resolution is significantly improved compared to what was reported previously and the newly described conformational states provide further mechanistic insight into the significance of protein movements that are relevant for DNA binding.

**Decision letter after peer review:**

Thank you for submitting your article "The dynamic nature of the human Origin Recognition Complex revealed through five cryoEM structures" for consideration by *eLife*. Your article has been reviewed by three peer reviewers, and the evaluation has been overseen by Sriram Subramaniam as the Reviewing Editor and John Kuriyan as the Senior Editor. The following individual involved in review of your submission has agreed to reveal their identity: Tim Grant (Reviewer #3).

The reviewers have discussed the reviews with one another and the Reviewing Editor has drafted this decision to help you prepare a revised submission.

Summary:

Jaremko et al. describe structures of multiple states of the human ORC complex, an essential replication complex in free and DNA-bound states. The structures contribute a major advance, as the resolution is significantly improved compared to the previous HsORC structure and describe conformational states that provide further mechanistic insight into HsORC function. Altogether these results are important and should be published.

Essential revisions:

1) Validation and analysis:

Supplementary file 1 and validation reports don't match. Supplementary file 1 lacks info.

– FSC (model-to-map) for final models should be provided.

– Maps should be displayed next to the model or a table indicating which conformation includes which subunits/domains/residues in the corresponding model should be provided to give the reader more clarity about which parts of the complex were modeled in each conformation and how the conformations differ from each other.

– Show EM maps for all ATPs sites, not only 1 conformation (in Figure 1—figure supplement 2).

2) Revision of Figures 1, 2, 4 and Video 1:

Figure 1:

– Sideview does not add much new info, rather show a cryoEM map next to the model to visualize which parts of the map have been modeled.

– Panel J) Show close-up of the β-341 hairpin loop of ORC_4O1AAA_ and show corresponding area in ScORC as well as cartoon of WHD domains from HsORC4 and ScORC4 side by side either in 1J or as part of Figure 4. The difference in this critical DNA binding area is a major finding of the manuscript and needs to be visualized for the reader.

Show intertwining of ORC2 and ORC3 from HsORC and comparison to DmORC.

Figure 2:

– Label subunits again. Panel 2B does not show structural change, needs a sideview or arrow to show the 90° tilt of the subunit in the bottom panel.

– Panel C-F: Display tilt angle of inactive versus autoinhibited conformation.

Figure 4:

– EM-Map hides DNA and it is not clear which residues interact with DNA. Show cartoon representation of structural models throughout this figure. Show top down view on dsDNA at appropriate zoom to visualize all interactions in 1 image.

Video 1:

– Use colorzone command or encircle WHD domain to highlight WHD domain unambiguously. Label red map, blue map and don't change color at the end of the video?

3) A deeper discussion of findings and the mechanistic model:

– The authors find shortening of an α-helix in the WHD domain of HsORC4 compared to ScORC that could account for sequence specificity of ScORC. Is there more data to support this? Which organisms have or don't have this helix? Does the presence of the helix correlate with sequence specificity of Orc? Are there prior sequencing data of corresponding ScORC4-WHD domain mutants that support this? This is an important finding, but is explored very little in the manuscript.

– The authors speculate that Arg89 of ORC3 could be involved in ATP coordination (subsection “Overall HsORC structure and comparison to ORC from *Saccharomyces cerevisiae* and *Drosophilamelanogaster*”). Ideally, they'd mutate the residue and test function. Absent that, is this residue conserved?

– The authors also speculate that HsORC may detect intracellular levels of ATP. Is there data to back this up? What is the Kd of ATP? Is it reasonable for acting as a sensor? If not, then this needs to be removed.

– The authors propose an interesting model for HsORC activation and DNA binding. Is there biochemical data in support of this model? Could comment a bit more on the role of the conserved WHD domain of ORC2 and the inactive state: Does the WHD domain in inactive state prevent nonspecific DNA binding of ORC2-5? Is there available DNA binding data on deletion mutants of WHD domain of ORC2?

– Discussion in regard to IDRs and phase separation: The text reads as if IDRs are present in several subunits – in which subunits and where are the IDRs? Are they predicted? Why are they not present in structures? Are they not resolved (likely owing to flexibility) or were they not included in expression constructs? Please clarify in discussion.

– Comment more on the characteristic structural changes from closed to open to DNA-bound (subunit to subunit translation, helical radius, etc). What does "open" refer to? Widening of the spiral?

Report on buried surface interaction area between ORC subunits to confirm exceptional intertwining of ORC2 and ORC3 from HsORC and DmORC.

– How do the authors come to the conclusion that the loop in ORC3 (Figure 1—figure supplement 5) is related to DNA binding, is this area close to DNA or DNA binding site? Is this area resolved in all conformations?

4) Multibody refinement:

– To solidify the ORC3·5 hinge, verify that hinge point is not an artefact of mask choice. Run multibody refine with a mask boundary between ORC1·4 and ORC2·3·5 and report on results.

– Figure legend is in discrepancy with text: Legend indicates that for multibody refinement particles from only 1 conformation were used, text states particles from 3 conformations were included.

– The text is also not very quantitative in description of the MBR.

5) DNA-bound structure:

It makes sense that the DNA sequence would not be in register because HsORC isn't sequence specific, but is the major/minor groove positioning in register? I would think that the positioning of the double helix is essentially fixed relative to the major/minor grooves. Can the authors make out DNA helix clearly? Clearer figures would help.

6) Sample preparation details:

In the sample preparation, addition of ATP in the buffer is mentioned. Were there any divalent cations, Mg^2+^ or Mn^2+^, included? What is the rationale for the ATP addition instead of ATPγS, AMPPNP or ADP·BFx? What are the green spheres shown next to the ATP molecules in Figure 1G-I? Why is ATP not hydrolyzed in ORC1·4 or ORC4·5?

7) In Figure 1—figure supplement 3D and E, the maps look of much lower resolution than 3.7 Å (ORC_OPEN_) or 4.3 Å (ORC_DNA_). Were these maps unsharpened? The authors may consider showing representative EM map regions or 3D-FSC.

8) DNA density:

For the ORCDNA complex, it will be important to show an EM map covering the DNA portion. Although base sequences are mixed, the DNA backbones should still be recognizable.

9) Resolution anisotropy:

The resolution is extremely directionally dependent, and the true resolution of the ORC open and ORC DNA maps is low in all but one direction. Based on the direction plots shown in Figure 1—figure supplement 3 all of the samples suffered somewhat from preferential orientation. I believe the authors should mention this, and perform a direction dependent resolution analysis of their maps (e.g. using the 3D FSC server – https://3dfsc.salk.edu/) and report the results.

---

## [Author Response]

Essential revisions:1) Validation and analysis:Supplementary file 1 and validation reports don't match. Supplementary file 1 lacks info.– FSC (model-to-map) for final models should be provided.

We have included the FSC (model-to-map) values at 0.143 and 0.5 in Supplementary file 1. Any other difference between the table and validation reports are due to the different validation software packages utilized. The statistics provided in Supplementary file 1 were generated from Phenix/Molprobity. Note – the structures have been further refined since the first submission and the geometry has greatly improved.

– Maps should be displayed next to the model or a table indicating which conformation includes which subunits/domains/residues in the corresponding model should be provided to give the reader more clarity about which parts of the complex were modeled in each conformation and how the conformations differ from each other.

We removed the sideview and added the density map next to each model in Figure 1. In addition, we have labelled each structure with their key feature (missing O1AAA+/O2WH, DNA present, open conformation) to better aid the reader.

– Show EM maps for all ATPs sites, not only 1 conformation (in Figure 1—figure supplement 2).

We added Figure 1—figure supplement 1 to include all the remaining ATP sites of all models with their corresponding EM densities.

2) Revision of Figures 1, 2, 4 and Video 1:Figure 1:– Sideview does not add much new info, rather show a cryoEM map next to the model to visualize which parts of the map have been modeled.

We removed the sideview and added the density map next to each model in Figure 1.

– Panel J) Show close-up of the β-341 hairpin loop of ORC_4O1AAA_ and show corresponding area in ScORC as well as cartoon of WHD domains from HsORC4 and ScORC4 side by side either in 1J or as part of Figure 4. The difference in this critical DNA binding area is a major finding of the manuscript and needs to be visualized for the reader.Show intertwining of ORC2 and ORC3 from HsORC and comparison to DmORC.

We added a figure of the ScORC4 loop in Figure 4D and moved the HsORC4 loop to Figure 4E with the overlaid density map removed for better visual comparison. We also show a superposition of the two loops in Figure 4F.

We included a figure of the ORC2-3 intertwine in our structure, ScORC, and DmORC for comparison in Figure 1—figure supplement 8, and calculated buried surface areas for a more quantitative comparison.

Figure 2:– Label subunits again. Panel 2B does not show structural change, needs a sideview or arrow to show the 90° tilt of the subunit in the bottom panel.

We replaced the maps in Figure 2A, B with models to better visualize the movement of O1AAA+ domain. The maps and process to generate the maps were moved to Figure 2—figure supplement 1 to better layout the method of visualizing the dynamic regions. We also changed Video 2 of the cryoEM map morphing to model morphing to better visualize the movement discussed in the text.

– Panel C-F: Display tilt angle of inactive versus autoinhibited conformation.

The change in conformation isn’t a simple tilt, but we have included Video 1 which clearly illustrates the different conformations of O2WH in the inactive and autoinhibited conformations.

Figure 4:– EM-Map hides DNA and it is not clear which residues interact with DNA. Show cartoon representation of structural models throughout this figure. Show top down view on dsDNA at appropriate zoom to visualize all interactions in 1 image.

We added a top down view of the DNA with important residues included in Figure 4A. We also included a figure of the DNA overlaid with the density map and a color-coded density map in Figure 4—figure supplement 1.

Video 1:– Use colorzone command or encircle WHD domain to highlight WHD domain unambiguously. Label red map, blue map and don't change color at the end of the video?

We added Video 1 of the ORC2 WHD that illustrates the movement of the domain more clearly. In addition, we added Figure 2—figure supplement 1 which lays out the low-pass filter method to visualize the ORC2 WHD and ORC1 AAA+ domain dynamics.

3) A deeper discussion of findings and the mechanistic model– The authors find shortening of an α-helix in the WHD domain of HsORC4 compared to ScORC that could account for sequence specificity of ScORC. Is there more data to support this? Which organisms have or don't have this helix? Does the presence of the helix correlate with sequence specificity of Orc? Are there prior sequencing data of corresponding ScORC4-WHD domain mutants that support this? This is an important finding, but is explored very little in the manuscript.

We removed the short comments from section 1, and expanded the discussion in the “DNA bound to HsORC” section.

We also added the really intriguing observation that the ORC4 helix insertions, along with sequence specificity, have likely evolved in *Saccharomyces cerevisae* budding yeast and other fungi coincident with loss of RNA-interference mediated silencing, while this region is absent in other organisms, including fungi, plants, and animals that do have RNAi. This is discussed in much greater detail in a paper that just came out in bioRxiv (Hu, et al., 2020) and that we now refer to in this manuscript.

– The authors speculate that Arg89 of ORC3 could be involved in ATP coordination (subsection “Overall HsORC structure and comparison to ORC from Saccharomyces cerevisiae and Drosophila melanogaster”). Ideally, they'd mutate the residue and test function. Absent that, is this residue conserved?

R89 is not conserved and after further inspection this claim has been removed from the text. However, we are confident in the remaining residues involvement in ATP coordination that are discussed.

– The authors also speculate that HsORC may detect intracellular levels of ATP. Is there data to back this up? What is the Kd of ATP? Is it reasonable for acting as a sensor? If not, then this needs to be removed.

We agree that this is probably too speculative and thus removed this from the text because we have no current data to validate the claim.

– The authors propose an interesting model for HsORC activation and DNA binding. Is there biochemical data in support of this model? Could comment a bit more on the role of the conserved WHD domain of ORC2 and the inactive state: Does the WHD domain in inactive state prevent nonspecific DNA binding of ORC2-5? Is there available DNA binding data on deletion mutants of WHD domain of ORC2?

As observed from the structures, the position of the ORC2 WHD in ORC2-5 occupies the same space as the DNA occupies in the ORC-DNA complex. Hence, we speculated that it obstructs DNA from binding at the central cavity in ORC2-5. ORC2-5 exhibits DNA binding activity (data not shown) but its exact mode of DNA interaction and how this interaction is related to the ORC-DNA conformation in the current study is still under investigation. Since metazoan ORC does not exhibit the sequence-specific DNA interaction observed in *S. cerevisiae*, we do not expect ORC2-5 or the ORC in the ORCDNA conformation to discriminate between specific and non-specific DNA-binding in our current study.

– Discussion in regard to IDRs and phase separation: The text reads as if IDRs are present in several subunits – in which subunits and where are the IDRs? Are they predicted? Why are they not present in structures? Are they not resolved (likely owing to flexibility) or were they not included in expression constructs? Please clarify in discussion.

We have further clarified the discussion of IDRs in the text: “These interactions between ORC1 and ORC2-5 may be facilitated by intrinsically disordered regions (IDRs) within the predicted N-terminal regions of ORC1 and ORC2 forming condensates via phase transitions (Parker, et al., 2019). Proteins containing IDRs are difficult to express and structurally resolve, therefore we removed the ORC1 N-terminal region from our construct and we were not able to resolve the ORC2 N-terminal region. Due to the absence of these regions, IDRs are likely not necessary for ORC assembly. “

– Comment more on the characteristic structural changes from closed to open to DNA-bound (subunit to subunit translation, helical radius, etc). What does "open" refer to? Widening of the spiral?Report on buried surface interaction area between ORC subunits to confirm exceptional intertwining of ORC2 and ORC3 from HsORC and DmORC.

We added to the discussion (subsection “Overall HsORC structure and comparison to ORC from *Saccharomyces cerevisiae* and *Drosophila melanogaster*”) describing widening of the spiral in the ORC_OPEN_ conformation and added Figure 3—figure supplement 2 which illustrates the change in conformation of a long α-helix in ORC that likely contributes to the stability of the open conformation.

In addition, we have utilized the ‘measure buriedarea’ function in ChimeraX to determine the solvent-accessible surface area buried between ORC subunits and determined that buried area between ORC2 and ORC3 (3296 Å^2^) is significantly larger than the next closest interacting subunits ORC4 and ORC5 (2707 Å^2^). This is discussed in the subsection “Overall HsORC structure and comparison to ORC from *Saccharomyces cerevisiae* and *Drosophila melanogaster*”. We have also added a figure in Figure 1—figure supplement 8 that focuses on the intertwine between O2 and O3.

– How do the authors come to the conclusion that the loop in ORC3 (Figure 1—figure supplement 5) is related to DNA binding, is this area close to DNA or DNA binding site? Is this area resolved in all conformations?

The cross-reference was incorrectly placed here. We also removed this information in the text.

4) Multibody refinement:– To solidify the ORC3·5 hinge, verify that hinge point is not an artefact of mask choice. Run multibody refine with a mask boundary between ORC1·4 and ORC2·3·5 and report on results.

To avoid mask bias, we performed Cryosparc 3D Variability to complement the Multibody Refinement results. Compared to Multibody Refinement, 3D Variability Analysis does not rely on user input to determine regions that move independently of one another. In addition, though ORC1 may be too small for a single body in Multibody Refinement, this is not the case for 3D Variability. Both analyses revealed the same prominent movements of the ORC_O1AAA_ particle population. We included the movements of both analyses in Video 3.

– Figure legend is in discrepancy with text: Legend indicates that for multibody refinement particles from only 1 conformation were used, text states particles from 3 conformations were included.– The text is also not very quantitative in description of the MBR.

We thank the reviewers for finding this mistake. There are 3 conformations and we have corrected this. The Multibody Refinement results for ORC_O2WH_ and ORC_OPEN_ are available in Figure 3—figure supplement 1, which shows the principle component analysis results. We have included text in the Figure 3 legend to clarify this.

In addition, we’ve added more text discussing the Multibody Refinement variance results and added Cryosparc’s 3D Variability Analysis to further validate the movements we observed (subsection “Hinge movement arising from the ORC3·5 interface”). We have added Video 3 that illustrates the movements generated from Multibody Refinement and 3D Variability Analysis.

5) DNA-bound structure:It makes sense that the DNA sequence would not be in register because HsORC isn't sequence specific, but is the major/minor groove positioning in register? I would think that the positioning of the double helix is essentially fixed relative to the major/minor grooves. Can the authors make out DNA helix clearly? Clearer figures would help.

We can roughly see the backbone but the resolution is too low. Even though we collected 2M particles we cannot resolve the various DNA conformations by 3D classification because of the number of particles remaining (see Figure 4—figure supplement 3). We therefore decided to report conservatively on our conclusions of the ORC-DNA binding. We have added a figure of the EM map covering the DNA in Figure 4—figure supplement 1 to better clarify the model and map of the DNA region.

6) Sample preparation details:In the sample preparation, addition of ATP in the buffer is mentioned. Were there any divalent cations, Mg^2+^ or Mn^2+^, included?

We did not include divalent cations in our purification buffers. However, the Mg^2+^ observed likely came from the lysate during purification. We included this information in the text (subsection “Overall HsORC structure and comparison to ORC from *Saccharomyces cerevisiae* and *Drosophila melanogaster*”) and Figure 1 legend.

What is the rationale for the ATP addition instead of ATPγS, AMPPNP or ADP·BFx?

Based on our previous study (Tocilj et al., 2017), we assumed ATP would engage ORC in an active state and we assumed this stabilized the complex. We attempted a data collection on an ORC sample that included ATPγS, but our best dataset was, in fact, with ATP.

What are the green spheres shown next to the ATP molecules in Figure 1G-I?

The green spheres are our interpretation of the density near the ATP and agrees with the positioning of Mg^2+^ as seen in many AAA+ ATPases. We have clarified this in the text (subsection “Overall HsORC structure and comparison to ORC from *Saccharomyces cerevisiae* and *Drosophila melanogaster*”) and Figure 1 legend.

Why is ATP not hydrolyzed in ORC1·4 or ORC4·5?

We presume that some of the ATP at the ORC1·4 site is hydrolyzed and the ORC_O2WH_ conformation is, in fact, the result of this hydrolysis, but not the others. It should be noted that these four conformations come from the same sample so some particles might have hydrolyzed ATP and some haven’t. The ATPase at the ORC4·5 interface is not functional based on our previous work (see Tocilj et al.), and would therefore not hydrolyze ATP. We also note, from work not yet published that the ATP is tightly bound at the ORC4·5 and is still clearly present from lysates even when ATP is not provided in the sample.

7) In Figure 1—figure supplement 3D and E, the maps look of much lower resolution than 3.7 Å (ORC_OPEN_) or 4.3 Å (ORC_DNA_). Were these maps unsharpened? The authors may consider showing representative EM map regions or 3D-FSC.

We also noticed the maps resolution appears lower than the reported values. Therefore, these maps were not sharpened. We performed 3DFSC and included plots in Figure 1—figure supplement 6.

8) DNA density:For the ORCDNA complex, it will be important to show an EM map covering the DNA portion. Although base sequences are mixed, the DNA backbones should still be recognizable.

The DNA backbone is somewhat recognizable. The resolution is low in this region due to multiple conformations of loosely bound DNA (Figure 4—figure supplement 3) and we cannot isolate individual conformations by 3D classification because the number of particles (<40k) in each class are too low to perform further classification.

We have added a figure of the EM map covering the DNA in Figure 4—figure supplement 1.

9) Resolution anisotropy:The resolution is extremely directionally dependent, and the true resolution of the ORC open and ORC DNA maps is low in all but one direction. Based on the direction plots shown in Figure 1—figure supplement 3 all of the samples suffered somewhat from preferential orientation. I believe the authors should mention this, and perform a direction dependent resolution analysis of their maps (e.g. using the 3D FSC server – https://3dfsc.salk.edu/) and report the results.

We performed 3DFSC and included plots in Figure 1—figure supplement 6. We also mentioned the analysis in the manuscript and referenced the work.